# A Simulation Benchmark for Autonomous Racing with Large-Scale Human Data

**Adrian Remonda**[1,2,4*] **Nicklas Hansen**[1] **Ayoub Raji**[3] **Nicola Musiu**[3]
**Marko Bertogna**[3] **Eduardo E. Veas**[2,4] **Xiaolong Wang**[1]

[1]UC San Diego  [2]TU-Graz  [3]Unimore  [4]Know-Center GmbH

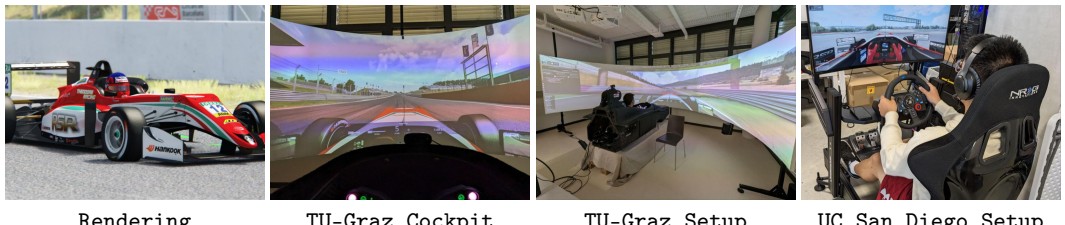

| Rendering | TU-Graz Cockpit | TU-Graz Setup | UC San Diego Setup |

*Figure 1:* **Overview.** We propose a high-fidelity racing simulation platform based on Assetto Corsa that enables reproducible algorithm benchmarking, as well as data collection with human drivers.

## Abstract

Despite the availability of international prize-money competitions, scaled vehicles, and simulation environments, research on autonomous racing and the control of sports cars operating close to the limit of handling has been limited by the high costs of vehicle acquisition and management, as well as the limited physics accuracy of open-source simulators. In this paper, we propose a racing simulation platform based on the simulator Assetto Corsa to test, validate, and benchmark autonomous driving algorithms, including reinforcement learning (RL) and classical Model Predictive Control (MPC), in realistic and challenging scenarios. Our contributions include the development of this simulation platform, several state-of-the-art algorithms tailored to the racing environment, and a comprehensive dataset collected from human drivers. Additionally, we evaluate algorithms in the offline RL setting. All the necessary code (including environment and benchmarks), working examples, datasets, and videos are publicly released and can be found at: https://assetto-corsa-gym.github.io.

## 1 Introduction

Autonomous driving has become an industry with different levels of applications affecting our daily lives, and it still has a large potential to continue revolutionizing future mobility and transportation. This paper explores a slightly different setting from day-to-day driving and focuses on driving an autonomous car at its physical limits, *i.e.*, autonomous racing. Specifically, the goal is to maneuver a car around a race track to achieve the lowest possible lap time and develop highly robust and generalizable models [Betz et al., 2022].

However, developing and testing algorithms for autonomous racing is a challenging and expensive task. Traditional testing methods, such as on-track testing, are limited in scope and can pose

---

*Work done during internship at UC San Diego. Correspondence can be directed to Adrian Remonda <aremonda@student.tugraz.at>.

38th Conference on Neural Information Processing Systems (NeurIPS 2024) Track on Datasets and Benchmarks.

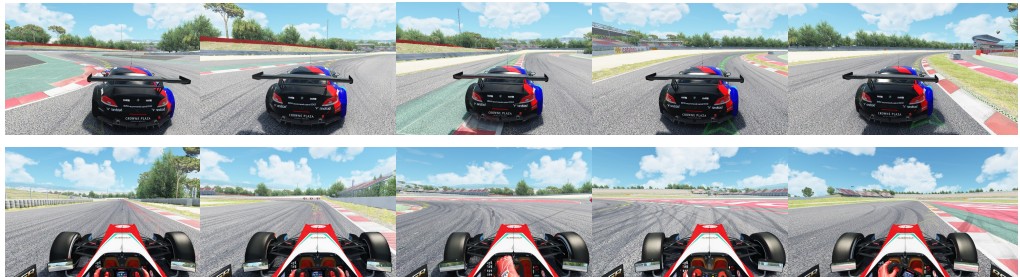

*Figure 2:* **Assetto Corsa.** A GT3 (*top*) and a F317 (*bottom*) car each turning a corner in the Assetto Corsa simulator. The simulator features a total of 178 official cars and 19 laser-scanned tracks, in addition to custom content created by the community. We develop a platform for interfacing with the simulator that can be used with both RL and MPC methods, as well as human drivers.

safety risks. In this context, simulations offer many advantages, such as the ability to replicate complex real-world scenarios, adjust environmental parameters, and collect large amounts of data for analysis. Crucially, it has been proven experience obtained from driving simulations can often be transferred to the real car. In fact, almost all Formula Racing teams have their own simulators for designing strategy and training racing drivers. If we can open source a platform for training autonomous agents, it will benefit both the research community and the racing industry. Currently, many simulators are available for autonomous driving research, such as Carla [Dosovitskiy et al., 2017] and F1Tenth [Babu and Behl, 2020], but none are specifically designed for high-speed racing. Wurman et al. [2022] have used Sony PlayStations for simulation in the Gran Turismo racing game, but the platform is presently not available to the community.

We present a novel, versatile, and realistic training and testing environment based on the high-fidelity racing simulator Assetto Corsa (visualized in Figure 2), which is widely used by professional drivers for practice. Our environment complies with the Gym interface and ROS2. Our framework leverages the plug-in interface provided by Assetto Corsa to obtain the real-time state of the vehicle and set controls. Assetto Corsa can be easily obtained and set up, making our platform accessible to a wider audience. This accessibility facilitates broader experimentation and development within the autonomous racing community.

Our framework supports the integration of Reinforcement Learning (RL) and control algorithms and features local and distributed execution capabilities. It can also simulate different weather conditions, opponent, tire wear, and fuel consumption scenarios. Additionally, our setup allows for recording human driving data, which is a key aspect of our research.

We include several state-of-the-art RL algorithms as well as classical control MPCs to benchmark autonomous racing in our platform. We present a comprehensive dataset that includes various cars and tracks. The dataset encompasses 64 million steps recorded while training one of our benchmark algorithms, Soft Actor-Critic [Haarnoja et al., 2018], as well as more than 900 laps from human drivers with different levels of expertise recorded in our simulator shown in Figure 1, ranging from a professional driver to beginners. This provides robust baselines for comparison. Finally, we demonstrate the usefulness of our dataset by providing insights and statistics on the collected data, along with empirical evidence of its utility in the RL setting, highlighting the value of human demonstrations in the field of self-driving racing. We will open-source code for the simulation environment, proposed evaluation platform, dataset, and baselines.

**Our key findings include:** *(i)* Human demonstrations provide a robust baseline for evaluating various model types; *(ii)* Better models can be achieved by using human demonstrations, as evidenced by improved lap times and sample efficiency; *(iii)* Utilizing demonstrations from different tracks enables rapid generalization to new tracks with fewer safety hazards, marking a significant advancement towards real-life racing deployment; and *(iv)* By bootstrapping with human demonstrations, we show that it is possible to drive without a reference path.

## 2  Related work

**Simulators for autonomous racing.** While there are multiple open-source simulators for autonomous driving [Kaur et al., 2021], among which CARLA [Dosovitskiy et al., 2017] is the most

complete and popular, there are limited platforms available for the racing domain [Babu and Behl, 2020, Balaji et al., 2020, Herman et al., 2021, Weiss and Behl, 2020, Wurman et al., 2022]. For example, environments for small-scale race cars have been proposed by Babu and Behl [2020] (F1Tenth) and Balaji et al. [2020] (DeepRacer), and Herman et al. [2021] (Learn-to-Race) proposes a simulator, framework, and dataset specifically tailored for the Roborace car and on only three tracks. The official F1 2017 racing game has been used by Weiss and Behl [2020] to build an end-to-end framework. Despite the availability of different tracks, the game lacks physics accuracy and realism, and the vehicles are limited to open-wheel F1 race cars. Professional Autonomous Racing teams build their simulation platforms upon the Unity engine and accurate multi-body models through Matlab/Simulink or dedicated Motorsport libraries [Betz et al., 2023, Raji et al., 2024]. The cost and complexity of modelling makes the knowledge of vehicle dynamics a strict requirement which limits the ease of access and use to the majority of the research community. Most similar to ours, Wurman et al. [2022] learns to drive in the Gran Turismo racing game using Soft Actor-Critic (SAC), and shows that a learned RL policy can outperform human players. However, the platform and methods used are not publicly available, making it difficult to build upon their research.

**Scientific research with Assetto Corsa (AC).** AC is an excellent platform to build upon for an autonomous racing environment, as it overcomes many of the weaknesses and gaps of aforementioned simulators. The game features 178 official cars and 19 laser-scanned tracks, in addition to custom content available online [Overtake.gg, Assetto Corsa Club]. AC can simulate different values of grip, weather conditions, and racing scenarios, such as single-vehicle performance laps, as well as online and offline multi-vehicle races. We leverage the plug-in interface of AC to control a vehicle in single-vehicle sessions and collect datasets. Our proposed platform offers similar capabilities to those proposed by Wurman et al. [2022], including support for distributed workers but without the need for PlayStation consoles for simulation. Previous literature has leveraged AC for various research problems, including road geometric research [de Frutos and Castro, 2021], deep learning for self-driving [Hilleli and El-Yaniv, 2016, Mahdavian and Martinez, 2018, Mentasti et al., 2020], racing game commentary [Ishigaki et al., 2021], and identification of driving styles [Vogel et al., 2022]. However, none of these works released interfaces nor code to the public.

**Algorithms for autonomous racing.** Typically, autonomous racing methods rely on the same conventional paradigm of perception, planning, and control [Betz et al., 2022] as in autonomous driving literature. Common approaches applied to real race cars adopt graph-based methods or smooth polynomial lane change for what concerns the online local planning [Raji et al., 2022, Stahl et al., 2019, Ticozzi et al., 2023]. The control often relies on classical optimization-based controllers such as Linear Quadratic Regulator (LQR) [Spisak et al., 2022] and Model Predictive Control (MPC) [Raji et al., 2023, Wischnewski et al., 2023]. Although not easily deployed on real cars, Reinforcement Learning (RL) has recently been used to learn highly performant control policies in simulation, rivaling professional human drivers [Remonda et al., 2022, Wurman et al., 2022]. Therefore, the availability of a complete autonomous racing simulation platform is crucial to make these methods generalizable and safe to use with real race cars. Our work presents such a platform, including a benchmark for RL methods, and a large-scale dataset of human drivers as well as RL replay data.

## 3 Preliminaries

**Problem definition.** We formulate autonomous racing as a Partially Observable Markov Decision Process (POMDP) [Bellman, 1957, Kaelbling et al., 1998] $\langle \mathcal{S}, \mathcal{A}, \mathcal{T}, r, \gamma \rangle$, where $\mathcal{S}$ denotes the state space, $\mathcal{A}$ is the action space, $\mathcal{T} \colon \mathcal{S} \times \mathcal{A} \mapsto \mathcal{S}$ is the (unknown) transition model, $r \colon \mathcal{S} \times \mathcal{A} \mapsto \mathbb{R}$ is a reward function, and $\gamma \in [0, 1)$ is the discount factor. Because the state $\mathbf{s}$ of the simulator cannot be observed directly, we approximate states as a sequence of the last $k$ observations (telemetry information) received from the environment. A deep Reinforcement Learning (RL) policy $\pi$ is trained to select actions $\mathbf{a}_t \sim \pi(\cdot|\mathbf{s}_t)$ at each time step $t$ such that the expected sum of discounted rewards $\mathbb{E}_\pi \left[ \sum_{t=0}^{\infty} \gamma^t r(\mathbf{s}_t, \mathbf{a}_t) \right]$ is maximized; the reward function thus serves as a proxy for lap time.

**Soft Actor-Critic (SAC).** Common model-free off-policy RL algorithms aim to estimate a state-action value function (critic) $Q \colon \mathcal{S} \times \mathcal{A} \mapsto \mathbb{R}$ by means of the single-step Bellman residual [Sutton, 1998] $\delta = Q_\theta(\mathbf{s}_t, \mathbf{a}_t) - \left( r(\mathbf{s}_t, \mathbf{a}_t) + \gamma \max_{\mathbf{a}_{t+1}} Q_\psi^{\text{tgt}}(\mathbf{s}_{t+1}, \mathbf{a}_t') \right)$. When the action space $\mathcal{A}$ is continuous, the second term – known as the *temporal difference* (TD) target – becomes intractable due to the $\max$ operator. To circumvent this, SAC [Haarnoja et al., 2018] additionally optimizes a stochas-

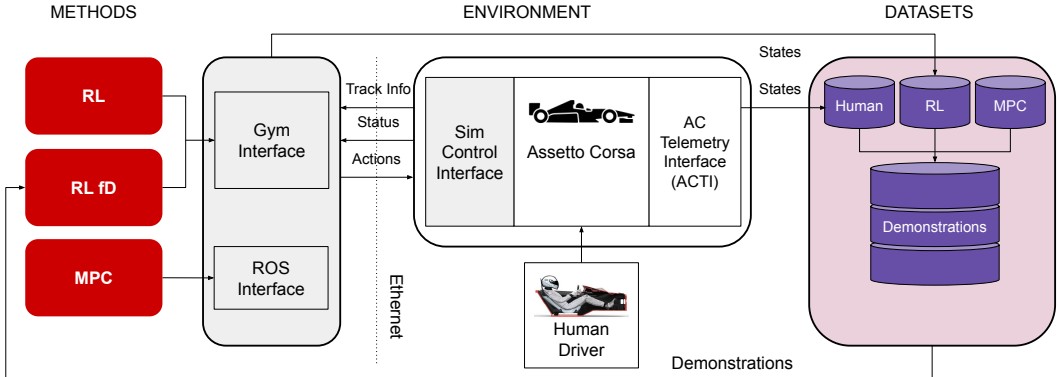

*Figure 3:* **Our proposed platform for autonomous racing.** We provide interfaces (*gray*) that *(1)* connect a simulator (Assetto Corsa) to autonomous racing methods, and *(2)* allow for human data collection. Interfaces receive track information and state, and execute actions in the simulator. Datasets (*purple*) are collected using an ACTI (Assetto Corsa Telemetry Interface) tool.

tic policy (actor) $\pi$ to maximize the value function via gradient ascent, regularized by a maximum entropy term: $\mathcal{L}_\pi(\theta) = \mathbb{E}_{\mathbf{s} \sim \mathcal{D}} \left[ Q(\mathbf{s}, \mathbf{a}) + \alpha \mathcal{H}(\pi(\cdot|\mathbf{s})) \right], \mathbf{a} \sim \pi(\cdot|\mathbf{s})$, where $\mathcal{H}$ is entropy and $\alpha$ is a temperature parameter balancing the two terms. While the single-step residual is the most commonly used objective for critic learning, we find it necessary for SAC to use multi-step ($n = 3$) residuals [Vecerik et al., 2017] during learning in the context of autonomous racing.

**TD-MPC2.** Model-based RL algorithm TD-MPC2 [Hansen et al., 2022, 2024] learns a latent decoder-free world model from sequential interaction data, and selects actions during inference by planning with the learned model. TD-MPC2 optimizes all components of the world model in an end-to-end manner using a combination of TD-learning (the single-step Bellman residual), reward prediction, and joint-embedding prediction [Grill et al., 2020]. During inference, TD-MPC2 follows the receding-horizon Model Predictive Control (MPC) framework and plans trajectories using a derivative-free (sampling-based) optimizer [Williams et al., 2015]. To accelerate planning, TD-MPC2 learns a model-free policy prior that is used to warm-start trajectory optimization.

## 4 A Simulation Benchmark for Autonomous Racing

We propose a racing simulation platform based on the Assetto Corsa simulator to test, validate, and benchmark algorithms for autonomous racing – including both RL and classical control – in realistic and challenging driving scenarios. In this section, we provide a technical overview of the design and features of our proposed platform, while deeper discussion of experiments is deferred to Section 5.

### 4.1 Platform Design

Our proposed platform, depicted in Figure 1, provides a simple and intuitive environment interface between autonomous racing algorithms (RL and MPC), human drivers, and high-fidelity racing simulation for which we leverage Assetto Corsa. Figure 3 provides an overview of this interface. At the center of the framework is the simulator, which should offer: *(i)* controls to setup and initiate the simulation, *(ii)* static information about track and vehicle (track borders, vehicle setup and characteristics), *(iii)* state of the simulation (*telemetry* data about dynamic parameters of the vehicle: rpm, speed, lateral and longitudinal accelerations, position in 3D, to name a few), and *(iv)* vehicle controls (minimally: steering, throttle, brake, gear shifts). The *Sim Control Interface* builds on the plug-in interface from Assetto Corsa (AC). This interface allows external applications to access telemetry data through a callback synchronized with the game's physics engine. AC with its physics engine operates at 300 Hz on a Windows-only platform. It runs only in real-time, which is a limitation for algorithms that can train faster than real-time, but also a strength for testing algorithms in real-world conditions. This architecture allows multiple instances of Assetto Corsa to run on different machines with a central node for data collection.

The Sim Control Interface relays information over Ethernet to the controller (Gym or ROS) in real time and it will deliver static information over TCP at a lower rate. Critically, we included in the plugin a feature that recovers the vehicle back to the track, essential for developing reinforcement learning algorithms. Status information and actions are streamed over UDP. The frequency of state updates can be set according to the frames-per-second of the game. For our experiments we used 25Hz, though the interface was tested up to 100 Hz. Actions in the simulation are applied using vJoy, a virtual joystick device recognized by the system as a standard joystick.

The controller can operate on either Linux or Windows, supporting both single and distributed systems. It connects to the Sim Control Interface, receiving simulation updates and providing an environment API as described in Section 4.2. When running a single system with the simulation, the controller applies actions directly using the vJoy instead of streaming them to the Sim Control Interface, to reduce latency. Additionally, the controller interface has all the information needed (settings, states, actions, rewards) to record demonstration datasets. When human drivers control the simulation, they directly operate on the simulator and the left-hand side of the diagram is not active. In this case, data is recorded using the Assetto Corsa Telemetry Interface [ACTI] which records data in MoTeC format, a standard format used in motorsports.

## 4.2 Environment Details

We provide a simple to use gym-compliant environment API for interfacing with RL algorithms. We detail observations, actions, reward function, and termination conditions in the following.

**Observations.** The environment is partially observable and agents asynchronously receive states $S \in \mathbb{R}^{125}$ with the most recent telemetry information provided by Assetto Corsa. To account for partial observability, states include telemetry data from the last *three* time steps, as well as absolute control values of the past two steps. Similar to previous work [Remonda et al., 2022], telemetry data includes: linear + angular velocities and accelerations of the car, a vector of range finder sensors (distance between track edge and car, if near), look-ahead curvature, force feedback from the steering wheel (human drivers rely on this to sense tire grip), angle between wheels and direction of movement, as well as 2D distance between the car and a reference path provided by AC. Notably, all cars are provided with the same reference path although the optimal path is vehicle-dependent; we can thus expect algorithms to benefit from deviating from this reference path to some extent.

**Actions.** The action space $\mathcal{A} \in \mathbb{R}^3$ is continuous and includes scalar controls for throttle, brake, and steering wheel, all normalized to $[-1, 1]$ for easy integration with RL algorithms. These values are translated to the maximum values allowed by the simulation. Crucially, we do not apply absolute controls but rather deltas to the current controls, which we find to greatly reduce oscillation. We use an automated gearbox for RL algorithms.

**Reward.** We opt for a reward function that is proportional to current forward velocity $v$, with a small penalty for deviations from the track axis: $r \doteq v \cdot (1 - \alpha d)$ where $d$ is $\ell_2$-distance to the reference path, and $\alpha$ is a constant coefficient that balances the weight of additional racing line supervision. Intuitively, increasing $\alpha$ allows the agent to deviate more from the reference path.

**Episode termination.** An episode terminates early if three or more wheels are outside of track boundaries, or when speed drops below 5 km/h for more than 2 seconds. Empirically, we find this to improve data efficiency significantly for RL algorithms that learn from scratch (*i.e.*, without access to human driving data). Humans are allowed to keep driving when termination conditions are met but the lap is invalidated.

## 4.3 Benchmark & Dataset

We consider a set of four different tracks and three cars for data collection and experimentation, which are all visualized in Figure 4. The tracks that we consider require diverse maneuvers: **Indianapolis (IND)**, an easy oval track; **Barcelona (BRN)**, with 14 distinct corners; **Austria (RBR)**, a balanced track with technical turns and high-speed straights; and **Monza (MNZ)**, the most challenging track with high-speed sections and complex chicanes. The cars that we consider all differ in setup, aerodynamics, and engine power (general vehicle dynamics): **Mazda Miata NA (Miata)**, top speed of 197 km/h; **Dallara F317 (F317)**, top speed of 250 km/h; and **BMW Z4 GT3 (GT3)**,

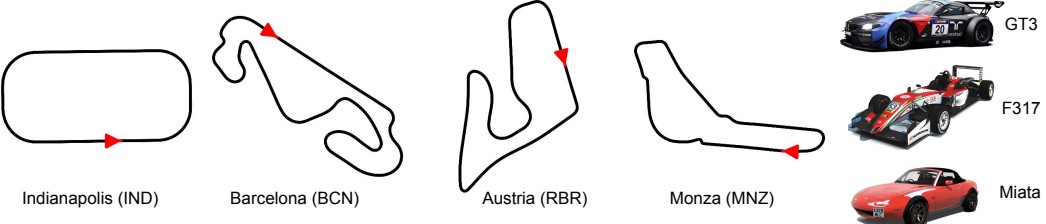

*Figure 4:* **Cars and tracks.** We consider a total of 4 different tracks (*left*), as well as 3 distinct cars (*right*). We collect and open-source human driving data for all tracks and cars considered.

top speed of 280 km/h. Our high diversity in tracks and cars ensures a more nuanced understanding of benchmarked algorithms.

Simulators were set up at University of California San Diego and Graz University of Technology, where a total of 15 human drivers were tasked with driving as fast as possible for at least 5 laps per track and car. Various categories of drivers participated: a professional e-sports driver, four experts who regularly train and compete online, five casual drivers with some experience, and five beginners using a racing simulator for the first time. Besides human driving data, we also collect a large dataset from the replay buffers of SAC policies trained from scratch. Table 1 summarizes our dataset.

*Table 1:* **Dataset.** We collect a total of 2.3M steps from human drivers of various skill levels, and 64M steps from SAC policies. A *stint* is a continuous period of driving without breaks.

| Car | Track | Stints | Laps | Steps | Steps |
|-----|-------|-------:|-----:|------:|------:|
| | | \multicolumn{3}{c}{**Human drivers**} | | **SAC** |
| F317 | BRN | 70 | 247 | 612,557 | 10M |
| F317 | MNZ | 19 | 117 | 288,582 | 10M |
| F317 | RBR | 24 | 142 | 295,679 | 10M |
| F317 | IND | 1 | 4 | 4,605 | 4M |
| GT3 | BRN | 37 | 181 | 501,206 | 10M |
| GT3 | RBR | 15 | 102 | 218,722 | 10M |
| GT3 | MNZ | 13 | 85 | 221,123 | 10M |
| Miata | BRN | 5 | 27 | 99,145 | 10M |
| Miata | MNZ | 2 | 10 | 38,395 | – |
| Miata | RBR | 3 | 12 | 32,971 | – |
| **Total** | | **189** | **927** | **2,312,985** | **64M** |

## 5  Experiments

The objective of our paper is to inspire further advancements in the field of autonomous racing. In this section, we validate the proposed environment and the dataset, and compare the performance of various algorithms against human drivers. Our experiments address key questions: the comparative performance of humans, RL algorithms, and classical MPCs; the benefits of incorporating human laps for training models; the impact of high-quality training data; the generalization of learning across different tracks; and the ability of agents to drive without predefined reference lines. All the necessary code (including environment and benchmarks) and working examples can be found at: https://github.com/dasGringuen/assetto_corsa_gym.

### 5.1  Methods

We consider 4 distinct methods for autonomous racing – built-in AI, MPC, SAC, and TD-MPC2 – as well as algorithmic variations of them. We summarize the methods as follows:

● **Built-in AI** controller provided by Assetto Corsa. We use it as-is.

● **Model Predictive Control** (MPC) [Raji et al., 2022, 2023], an optimization-based controller that utilizes a single-track vehicle model. The approach requires substantial domain knowledge and engineering efforts for each track and car. This is the only method presented that has been directly deployed to a real race car.

● **Soft Actor-Critic** (SAC) [Haarnoja et al., 2018], a model-free RL algorithm. We train SAC via online interaction by default, *i.e.*, initialized from scratch without any prior data. Our SAC implementation uses $n$-step returns ($n = 3$) which we found to be critical for learning.

● **TD-MPC2** [Hansen et al., 2024], a model-based RL algorithm. A key strength of TD-MPC2 is its ability to consume various data sources: human driving data, existing data collected via RL, as well as its own online interaction data, either from a single race track or across multiple tracks.

*Table 2:* **Best lap times (s).** Best result achieved by each method for every track and car (↓ lower is better). SAC fD, which uses human demonstrations, is generally on par with the best human drivers using the F317 car. Human experts are slightly faster with the GT3 car.

| Car | Track | Built-in AI | Best Human | SAC | SAC fD | TD-MPC2 fD | TD-MPC2 ft | MPC | IQL |
|-----|-------|-------------|------------|------|--------|------------|------------|------|------|
| F317 | IND | 59.88 | **59.38** | 59.84 | 59.85 | 60.10 | 59.80 | 59.41 | 59.81 |
| | BRN | 99.74 | **97.52** | 97.67 | 97.54 | 98.27 | 98.27 | 100.54 | 98.62 |
| | RBR | 84.85 | 83.53 | **83.51** | 83.61 | 84.69 | 84.50 | 86.70 | 84.80 |
| | MNZ | **Fail** | 104.31 | 104.41 | **104.08** | 105.90 | 105.36 | 106.32 | 104.64 |
| GT3 | BRN | 109.66 | **108.38** | 109.51 | 108.96 | - | - | 112.22 | 111.51 |
| | RBR | 91.56 | **91.16** | 93.11 | 92.02 | - | - | 95.00 | 93.68 |
| | MNZ | 111.44 | **110.87** | 112.69 | 111.64 | - | - | 112.66 | **Fail** |
| Miata | BRN | 153.62 | 158.28 | **152.12** | - | - | - | - | - |

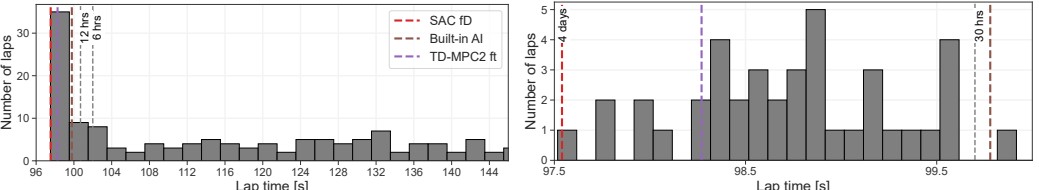

*Figure 6:* **Lap times of human drivers on BRN with a F317 car.** ↓ Lower is better. *(left)* Overall distribution; *(right)* zoomed-in view of the best laps. Vertical lines indicate best laps by each method at end of training. SAC and TD-MPC2 outperform most human drivers with just 6 hours of training.

• **IQL** [Kostrikov et al., 2021], an offline RL algorithm learning from offline datasets, suited for scenarios where online exploration is infeasible. We report results without finetuning.

**Method variations.** We evaluate performance of the data-driven SAC and TD-MPC2 methods both when learning from scratch (no prior data), as well as with various additional sources of data; we use a **fD** suffix to denote variants in which an algorithm is initialized with a pre-existing (offline) dataset of human driving. When learning from both offline and online interaction data, we choose to maintain separate buffers for each data source and sample such that each mini-batch contains 50% data from each source [Feng et al., 2023]. We also consider a setting where TD-MPC2 is pretrained on one track and subsequently finetuned on a different track, retaining both model parameters and data from the pretraining phase; we denote this **TD-MPC2 ft**. We omit SAC ft due to poor performance.

**Training and evaluation.** We trained the RL algorithms over multiple episodes each one consisting on multiple laps on the track until 15,000 steps were reached (about 7 laps depending on track and car). All experiments were conducted on workstations equipped with NVIDIA GeForce RTX 4090 GPUs. We report the best lap time achieved by each agent after 4 days of training.

### 5.2  Results

**Q1: How good are algorithms and humans at racing?** Table 2 shows the best lap times achieved by humans and algorithms on each track and car, and Figure 5 shows average lap time of algorithms relative to the *best* human lap. We observe that with the F317 car, SAC-fD is slightly better than the human pro driver, while in the GT3, human experts (including the pro driver) are slightly faster than SAC-fD. The

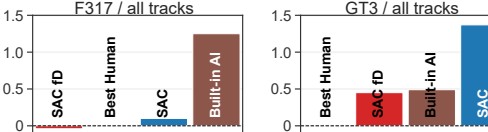

*Figure 5:* **Mean delta lap time vs. best human.** Across all 4 tracks. SAC fD performs better than the best human with F317 but worse with GT3.

built-in AI was typically slower and failed to complete a lap in one instance. Our MPC method was the best model in INDI/F317 and, although on average slower by a large margin, it can more readily be applied to racing on a real car. Figure 10 shows the lap time distribution of human drivers. Gray vertical lines indicate the wall time needed by SAC and TD-MPC2 to be faster than these human laps. Both outperform most human drivers after just 6 hours of training. SAC seeds approx. 4 days of training to match the *best* human driver.

**Q2: Does pre-training on data from multiple tracks enable few-shot transfer to unseen tracks?** A key benefit of data-driven approaches (such as RL) is their potential for learning *generalizable* racing policies that transfer across tasks and cars. In the following, we investigate whether pretraining TD-MPC2 on a set of (source) tracks and then finetuning on a target track (MNZ and AT in our experiments) improves data-efficiency and/or reduces number of crashes on the new track. Specifically, we first pretrain TD-MPC2 on human driving data from all tracks (excluding the two target tracks), as well as online interaction data on the BRN track. Next, we finetune this TD-MPC2 model on the two target tracks for 4 days each (separate finetuning procedures) and compare its lap time against TD-MPC2 and SAC with human driving data only from the target task. We find that it is helpful to differentiate tracks by including an additional track ID in observations during both pretraining and finetuning.

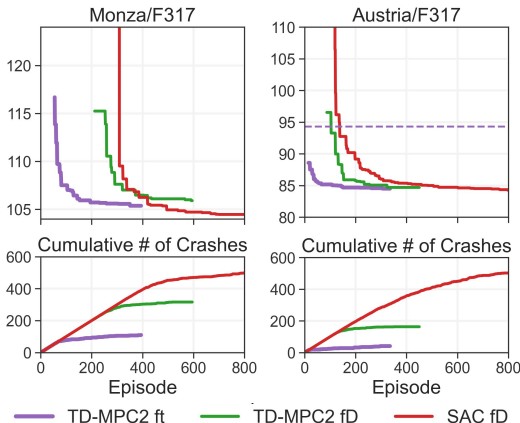

*Figure 7:* **Few-shot transfer.** Finetuned TD-MPC2 (*ft*) on target tracks vs SAC fD and TD-MPC2 fD. *(top)* Lap time vs. episodes. Each episode is about 7 laps unless the car crashes. Due to training for a fixed number of steps, the plots show curves of different lengths. *(bottom)* Cumulative number of times an agent did not finish.

Results are shown in Figure 7 (*top*). TD-MPC2 exhibits faster convergence when pretrained on other tracks, and completes full laps on the target tracks after only a few episodes whereas SAC fD and TD-MPC2 fD require hundreds of episodes. The benefit of pretraining is also clearly reflected in the cumulative number of crashes during training (*bottom*); TD-MPC2 crashes far less frequently when pretrained, and TD-MPC2 fD similarly crashes less frequently than SAC fD. We omit pretraining and finetuning results using SAC as we find it to perform poorly in this setting.

**Q3: Does better (faster) human demonstrations improve RL performance?** We hypothesize that data from expert human drivers will be more beneficial to RL algorithms than that of less experienced drivers. To test this hypothesis, we split human driving data into

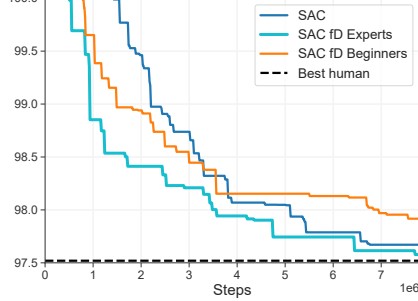

*Figure 8:* **Demonstration quality.** Lap time (s) of SAC trained with human data of varying experience level on BRN/F317. Better human data improves convergence considerably.

two categories: experts (including the pro driver), and beginners (including intermediate drivers). Then, we train SAC on BRN with a F317 using data from each group; results are shown in Figure 8. Our results indicate that having high-quality demonstrations do indeed lead to faster convergence vs. provided with slower human lap data as well as when training from scratch. However, we find that RL algorithms still benefit from beginner demonstrations substantially early in training, but result in overall slower lap times at later stages of training. We conjecture that this is because the RL algorithm becomes too biased towards human driving behavior.

**Q4: Can RL algorithms learn to race without a predefined reference path?** Our previous experiments use a reward function that mildly penalizes deviation from a predefined reference path provided by Assetto Corsa. Providing this reference helps with exploration (especially early in training) but is not optimal. In general, calculating the optimal path is very challenging as it depends not only on track geometry but also on vehicle dynamics [Cardamone et al., 2010]. We now investigate whether RL algorithms can succeed without such a reference path; results are shown in Table 3.

*Table 3:* **Lap times without reference path.** SAC fails to complete a lap without a reference path, whereas SAC fD completes laps albeit slightly slower.

| Method / Car | F317 | GT3 |
|---|---|---|
| SAC | 97.67 | 109.51 |
| SAC, no ref | **Fail** | **Fail** |
| SAC fD, no ref | 99.80 | 110.98 |

Interestingly, SAC trained from scratch fails to complete a lap without the reference path, whereas it succeeds when provided with human demonstrations (albeit still slightly slower than when provided with a reference path). This strongly suggests that both reference paths and human driving data help overcome the challenges of exploration. We acknowledge that Wurman et al. [2022], too, has successfully trained an autonomous racing policy without a reference path. However, our setting is more general as their approach assumes access to the exact position of the car on a given track, whereas our setting only relies on range finder sensors and thus has potential to generalize across tracks.

## 6 Conclusion

Our results validate the quality of both the simulator and the collected dataset, as well as some of their potential use cases in future research. In particular, we demonstrate that data-driven methods benefit greatly from human demonstrations, and that existing algorithms can be used for few-shot transfer to unseen tracks. We provide source code for the environment, benchmark, and baseline algorithms, and we look forward to seeing what the learning, control, and autonomous racing communities will use it for.

**Limitations.** We acknowledge that AC runs only in real-time, which can be a limitation for algorithms capable of faster-than-real-time training. However, this real-time constraint is beneficial for testing algorithms in real-world conditions. This limitation can be overcome by running many workers in parallel to collect data. We do not experiment with image observations in this work, but acknowledge that it would be an interesting direction for future research. In particular, driving from raw image observations rather than, *e.g.*, range finder sensors, would more closely resemble the sensory information that human drivers can access. Our RL policies are in some cases outperformed by experts, indicating that the maximum potential performance has not yet been achieved even on existing tracks and cars. One path for improvement is addressing the suboptimal autoshifting mechanism in AC. Allowing an RL agent to control the shifts is challenging due to the mix of continuous and discrete action spaces, but it holds potential for enhanced performance.

**Potential negative societal impact.** The advent of autonomous vehicles presents significant challenges, including the displacement of traditional labor roles and potential ethical dilemmas surrounding AI decision-making in high-risk scenarios. Additionally, public trust in autonomous technology could be jeopardized by accidents, complicating the regulatory landscape and raising complex liability issues.

## 7 Acknowledgments

We extend our gratitude to the current and former members of Unimore Racing who contributed to this platform, particularly Francesco Moretti, Andrea Serafini, and Francesco Gatti. Special thanks to Aleksandra Krajnc and Haichuan Che for their invaluable assistance with data collection. This project was supported, in part, by the Amazon Research Award, the Intel Rising Star Faculty Award, the NVIDIA Graduate Fellowship, gifts from Qualcomm, and Know-Center GmbH. Know-Center is funded within the Austrian COMET Program.

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

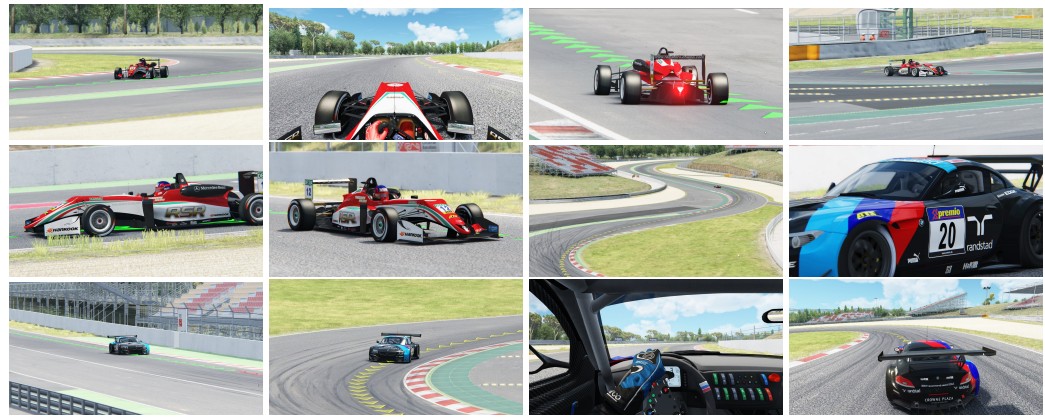

*Figure 9:* **Assetto Corsa render.** Different views of the cars in Assetto Corsa

## A  Dataset access

**Web with more results and videos:** can be accessed via `https://assetto-corsa-gym.github.io/`.

**Source code:** can be accessed through `https://github.com/dasGringuen/assetto_corsa_gym`.

**URL and data cards:** The dataset can be viewed at `https://huggingface.co/datasets/dasgringuen/assettoCorsaGym`.

**Author statement:** We bear all responsibility in case of violation of rights. We confirm the CC BY (Attribution) 4.0 license for this dataset.

**Hosting, licensing, and maintenance plan:** We host the dataset on HuggingFace, and we confirm that we will provide the necessary maintenance for this dataset.

**Structured metadata:** The metadata is available at `https://huggingface.co/datasets/dasgringuen/assettoCorsaGym` and `https://github.com/dasGringuen/assetto_corsa_gym/tree/main/data`.

## B  Accessibility and Usage Rights of Assetto Corsa

The Assetto Corsa software is readily accessible and affordable, available for purchase on Steam. However, for those intending to publish research work or use the software for commercial purposes, it is necessary to contact Kunos Simulazioni (the developer of Assetto Corsa) (`kunos-simulazioni.com`) to obtain the appropriate permissions.

## C  Illustrations

Figure 9 shows different views of the cars in Assetto Corsa.

## D  Implementation Details - MPC

The classical MPC used in this work is based on Raji et al. [2023]. The solution considers a single-track model written in curvilinear coordinates, including a simplified Pacejka Magic Formula for the tire model, aerodynamic effects, combined effects, and longitudinal load transfer. The parameters of the model are identified using experimental data gathered by manually driving the vehicle and using information included in the configuration files provided by AC for each car.

The state vector is defined as $\tilde{x} = [s; n; \mu; v_x; v_y; r; \delta; D]$, and the input vector as $\tilde{u} = [\Delta\delta; \Delta D]$,, where $s$ is the progress along the path, $n$ is the orthogonal deviation from the path, and $\mu$ is the local heading. $v_x$ and $v_y$ are the longitudinal and lateral velocities, respectively. The yaw rate of the vehicle is represented by $r$. $\delta$ represents the steering angle, and $D$ is the desired longitudinal acceleration, which is converted into throttle and brake commands through a low-level controller based on a feedforward and PI controller. $\Delta\delta$ and $\Delta D$ represent the derivatives of the commands and are used as inputs in the optimization problem.

The cost function is formulated as:

$$J_{MPC}(x_t, u_t) = q_s s_t^2 + q_n n_t^2 + q_\mu \mu_t^2 + q_v v_t^2 + q_r + u^T R u + B(x_t), \tag{1}$$

where $q_n$ and $q_\mu$ are path-following weights, and $q_s$ is a weight on the progression along the path. The regularization term $B(x_k) = q_B \alpha_r^2$ penalizes the rear slip angle, while $u^T R u$ is a regularizer on the input rates where $R$ is a diagonal weight matrix.

The MPC problem is formulated as

$$\min_{X,U} \sum_{t=0}^{T} J_{MPC}(x_t, u_t) \tag{2a}$$

$$s.t. \ x_0 = \hat{x}, \tag{2b}$$

$$x_{t+1} = f_t^d(x_t, u_t), \tag{2c}$$

$$x_t \in X_{track} \quad x_t \in X_{ellipse}, \tag{2d}$$

$$a_t \in \boldsymbol{A}, \ u_t \in \boldsymbol{U}, t = 0, \ldots, T. \tag{2e}$$

where $X = [x_0, ..., x_T]$, and $U = [u_0, ..., u_T]$ are the states and inputs over the horizon. $X_{ellipse}$ represents a friction ellipse constraint on the lateral and combined tire forces, and $X_{track}$ constrains the lateral deviation $n$, ensuring that the vehicle stays on the track. $\hat{x}$ is the current curvilinear state, and $T$ is the prediction horizon. $\boldsymbol{A}$ and $\boldsymbol{U}$ are box constraints for the physical inputs $a = [\delta; D]$ and their rate of change $u$.

The horizon, over which the optimization is performed, is composed of 63 steps of 40 ms, creating a prediction of 2.5 seconds. The model is discretized in time at each step $x_{t+1}$ using a Runge-Kutta fourth-order integrator. The controller is executed at a rate of 100 Hz, respecting a maximum total computational time of 10 ms.

Compared to the original work [Raji et al., 2023], the main differences are:

- Instead of a tricycle model where the yaw moment generated by the rear wheels is considered, we simplified the model to a single-track model as in [Raji et al., 2022]. This choice was made because the racecars used in this work do not exhibit any particular behavior produced by the rear mechanical differential; therefore, the dynamics can be modeled accurately enough with a more classical single-track model.

- The trajectory given to the MPC as a target to be tracked, composed of a path and a speed profile, is generated by an optimization problem executed offline, which uses the path provided by AC as a reference and aims to minimize the lap time. The cost of following the reference path was properly weighted to speed up the time needed to find a feasible solution. In this way, the trajectory produced consists of a path almost identical to the one provided by AC but with an optimized speed profile.

- Once the controller was able to satisfactorily track the path and speed profile, the weight on the progression along the path $q_s$ was increased progressively to reduce the lap time, similar to what was done in [Liniger et al., 2015].

We believe that the gap between the MPC and the best laps produced in this work is mainly caused by the fact that the trajectory does not exploit the entire width of the track and that the AC reference is not always optimal. Additionally, the low-level velocity controller does not always track the target accurately enough, causing the car to exceed the desired velocity at the turn entrance. This creates a gap between the MPC prediction and the real maneuver produced by the actuated longitudinal command.

# E  Implementation Details - Reinforcement Learning

## E.1  SAC

We base our SAC [Haarnoja et al., 2018] code on the implementation from `https://github.com/toshikwa/discor.pytorch`, making minor modifications and tuning hyperparameters. We used the hyperparameters listed in Table 4. We trained each model for 4 days, approximately 2 million steps per day.

*Table 4:* **SAC hyperparameters.** We used the same hyperparameters across all tracks and cars.

| Hyperparameter | Value |
|---|---|
| Batch size | 128 |
| Memory size | 10,000,000 |
| Offline buffer size | 10,000,000 |
| Update interval | 1 |
| Start steps | 2000 |
| Evaluation interval | 100,000 |
| Number of evaluation episodes | 1 |
| Checkpoint frequency | 200,000 |
| Discount factor ($\gamma$) | 0.992 |
| N-step | 3 |
| Policy learning rate | 0.0003 |
| Q-function learning rate | 0.0003 |
| Entropy learning rate | 0.0003 |
| Policy hidden units | [256, 256, 256] |
| Q-function hidden units | [256, 256, 256] |
| Target update coefficient | 0.005 |
| Number of Q functions | 2 |

## E.2  TD-MPC2

We used the official implementation from Hansen et al. [2024] which can be found at `https://github.com/nicklashansen/tdmpc2`, making minor modifications and tuning a few hyperparameters. To fit the real-time windows, we disabled planning. Additionally, we collected experiences and only updated the model parameters at the end of each episode. Due to hardware limitations, without this adjustment, we found that the model failed to train properly when the car was moving at higher speeds. We train each model for 4 days to approximately 1 million steps per day.

**Hyperparameters.** We used the same hyperparameters which are listed in Table 5. Our adapter parameters are in bold. Specifically, we set the discount factor $\gamma$ to a fixed value of 0.992, as the heuristic used in the original TM-MPC2 paper was not applicable to our tasks due to our longer horizon of 15k steps per episode. We keep the default configuration of 5 million parameters.

*Table 5:* **TD-MPC2 hyperparameters.** We use the same hyperparameters across all tracks and cars.

| Hyperparameter | Value |
|---|---|
| **Planning** | |
| Horizon ($H$) | 3 |
| Iterations | 6 |
| Population size | 512 |
| Policy prior samples | 24 |
| Number of elites | 64 |
| Minimum std. | 0.05 |
| Maximum std. | 2 |
| Temperature | 0.5 |
| Momentum | No |
| | |
| **Policy prior** | |
| Log std. min. | **-20** |
| Log std. max. | 2 |
| | |
| **Replay buffer** | |
| Capacity | **10,000,000** |
| Sampling | Uniform |
| | |
| **Architecture (5M)** | |
| Encoder dim | 256 |
| MLP dim | 512 |
| Latent state dim | 512 |
| Task embedding dim | 96 |
| Task embedding norm | 1 |
| Activation | LayerNorm + Mish |
| $Q$-function dropout rate | 1% |
| Number of $Q$-functions | 5 |
| Number of reward/value bins | 101 |
| SimNorm dim ($V$) | 8 |
| SimNorm temperature ($\tau$) | 1 |
| | |
| **Optimization** | |
| Update-to-data ratio | 1 |
| Batch size | 256 |
| Joint-embedding coef. | 20 |
| Reward prediction coef. | 0.1 |
| Value prediction coef. | 0.1 |
| Temporal coef. ($\lambda$) | 0.5 |
| $Q$-fn. momentum coef. | 0.99 |
| Policy prior entropy coef. | $1 \times 10^{-4}$ |
| Policy prior loss norm. | Moving $(5\%, 95\%)$ percentiles |
| Optimizer | Adam |
| Learning rate | $3 \times 10^{-4}$ |
| Encoder learning rate | $1 \times 10^{-4}$ |
| Gradient clip norm | 20 |
| Discount factor | **0.992** |
| Seed steps | **2000** |

*Table 6:* **IQL Hyperparameters.** These values are consistent across all tracks and cars.

| Hyperparameter | Value |
| --- | --- |
| Evaluation interval | 50,000 steps |
| Batch size | 256 |
| Number of pretraining steps | $1 \times 10^6$ |
| Replay buffer size | **10,000,000** |
| Actor learning rate | $3 \times 10^{-4}$ |
| Value learning rate | $3 \times 10^{-4}$ |
| Critic learning rate | $3 \times 10^{-4}$ |
| Discount factor | **0.992** |
| Hidden layer dimensions | (256, 256, 256) |
| Temperature | **1** |
| Expectile ($\tau$) | **0.6** |

### E.3 IQL

We used the official implementation from Kostrikov et al. [2021] available at `https://github.com/ikostrikov/implicit_q_learning`, with minor modifications and hyperparameter tuning. This implementation uses JAX, which lacks CUDA support on Windows. Therefore, we train and evaluate the model on a Linux system while connected via Ethernet to a Windows machine running Assetto Corsa. All evaluations are conducted in the offline setting (no fine-tuning), and we assess performance every 50,000 training steps.

**Hyperparameters.** We follow the original paper's hyperparameters, tuning only the temperature and expectile ($\tau$) parameters as listed in Table 6. Custom parameters are highlighted in bold.

# F  Environment

## F.1  Settings

For all our experiments, we set the $\alpha$ coefficient to $\frac{1}{12}$. $\alpha$ balances the weight of the racing line supervision. The actions are applied at a frequency of 25 Hz and are applied locally, meaning they are executed directly from the gym part of the framework without sending them back through UDP. We use relative actions in all experiments.

The plugin code was built around the API and the shared memory structures offered by AC. Most channels are available through shared memory, which are updated by the game at the same frequency as the physics engine. The Assetto Corsa plugin API documentation, which details additional functions, can be found in the official documentation at https://assettocorsamods.net/attachments/acpythondocumentation-pdf.1135/, and the shared memory documentation can be found at https://assettocorsamods.net/threads/doc-shared-memory-reference-acc.3061/.

## F.2  Vehicle Specifications

We consider a total of three cars, each differing in top speed and steerability:

**Mazda MX5 NA:** The Mazda MX5 NA has an all-steel body shell and a light-weight aluminium hood.

**BMW Z4 GT3:** The BMW Z4 GT3 is the race car version of the Z4. This is the most realistic and challenging vehicle we consider. It can reach speeds of up to 280 km/h with double the weight of the Dallara, making it a formidable test for our models and human drivers.

**Formula 3:** Formula 3 car in line with FIA Formula 3 regulations. This is of mid difficulty with a top speed of approximately 250 km/h. This model is contributed by the community (https://www.racedepartment.com/downloads/rsr-formula-3.8040/).

*Table 7:* **Specifications of Mazda MX5 NA, BMW Z4 GT3, and Formula F317.**

| Specification | Mazda MX5 NA | BMW Z4 GT3 | Formula 3 |
|---|---|---|---|
| Weight | 1040 kg | 1265 kg | 550 kg |
| Power | 130 bhp | 530 bhp | 246 bhp |
| Torque | 152 Nm | 520 Nm | 320 Nm |
| Top Speed | 197+ km/h | 280+ km/h | 230+ km/h |
| Acceleration (0-100 km/h) | 8.8 s | – s | – s |
| Weight/Power Ratio | 8 kg/hp | 2.38 kg/hp | 2.03 kg/bhp |

## F.3 Observations used by the RL algorithms

Table 8 shows the observations used by the RL algorithms, selected from the available inputs.

*Table 8:* **Used channels by the RL algorithms**

| Input | Description |
|---|---|
| Speed | The overall speed of the car, measured in kilometers per hour (km/h). |
| FFB | Steering wheel force feedback. Torque provided by the steering wheel to replicate steering wheel movements. In a real car, turning the steering wheel causes it to naturally pull back to the center. Force feedback (FFB) replicates this sensation, allowing drivers to feel the car's grip. |
| Engine RPM | The revolutions per minute of the engine. |
| AccelX, Y | The acceleration of the car along the X (longitudinal) and Y (lateral) axes (m/s²). |
| ActualGear | The current gear selected in the car's transmission system. |
| Angular_velocity_y | The rate of rotation around the car's vertical axis (yaw rate) (radians per second). |
| Local_velocity_x | The velocity of the car in the forward direction relative to the car's own frame of reference. |
| Local_velocity_y | The velocity of the car in the sideways direction relative to the car's own frame of reference. |
| SlipAngle_fl | Angle between the direction the front left wheel is pointing and the direction it is actually moving. This is crucial for understanding tire grip and potential understeer or oversteer. |
| SlipAngle_fr | Slip angle front right wheel. |
| SlipAngle_rl | Slip angle rear left wheel. Important for understanding rear tire grip and the car's stability. |
| SlipAngle_rr | Slip angle for the rear right wheel |
| look-ahead curvature | The curvature of the path ahead of the car. |
| 2D distance to reference path | The 2D distance between the car and a reference path. We used the reference path one provided by AC. |
| Range finder sensors | Vector of range finder sensors, indicating the distance between the track edge and the car, if near. |

## F.4 Available inputs

Table 9 shows the telemetry inputs send by the *Sim Control interface*.

*Table 9:* Telemetry channels sent by the *Sim Control interface*

| Input | Description |
|---|---|
| steps | Tracks the number of steps in the update loop. |
| packetId | Index of the shared memory's current step. |
| currentTime | Current in-game time. |
| world_position_x | X-coordinate of the car's position in the world. |
| world_position_y | Y-coordinate of the car's position in the world. |
| world_position_z | Z-coordinate of the car's position in the world. |
| speed | Car's speed in meters per second. |
| local_velocity_x | Car's local velocity in the X direction. |
| local_velocity_y | Car's local velocity in the Y direction. |
| local_velocity_z | Car's local velocity in the Z direction. |
| accelX | Acceleration in the X direction in G-forces. |
| accelY | Acceleration in the Y direction in G-forces. |
| steerAngle | Angle of steer. In degrees. |
| accStatus | Value of gas pedal: 0 to 1 (fully pressed). |
| brakeStatus | Value of brake pedal: 0 to 1 (fully pressed). |
| actualGear | Selected gear (0 is reverse, 1 is neutral, 2 is first gear). |
| RPM | Engine revolutions per minute. |
| yaw | Heading of the car on world coordinates. |
| pitch | Pitch of the car on world coordinates. |
| roll | Roll of the car on world coordinates. |
| LastFF | Last force feedback signal sent to the wheel. |
| NormalizedSplinePosition | Position of the car on the track in normalized [0,1]. |
| LapCount | Number of completed laps by the player. |
| BestLap | Best lap time in seconds. |
| completedLaps | Number of laps completed. |
| numberOfTyresOut | How many tires are out of the track |
| angular_velocity_x | Angular velocity around the X-axis. |
| angular_velocity_y | Angular velocity around the Y-axis. |
| angular_velocity_z | Angular velocity around the Z-axis. |
| velocity_x | Car's velocity in the X direction. |
| velocity_y | Car's velocity in the Y direction. |
| velocity_z | Car's velocity in the Z direction. |
| cgHeight | Height of the center of gravity of the car from the ground. |
| SlipAngle_fl | Slip angle of the front-left tire. |
| SlipAngle_fr | Slip angle of the front-right tire. |
| SlipAngle_rl | Slip angle of the rear-left tire. |
| SlipAngle_rr | Slip angle of the rear-right tire. |
| tyre_heading_vector | tire Heading Vector. |
| tyre_slip_ratio_fl | Slip ratio of the front-left tire. |
| tyre_slip_ratio_fr | Slip ratio of the front-right tire. |
| tyre_slip_ratio_rl | Slip ratio of the rear-left tire. |
| tyre_slip_ratio_rr | Slip ratio of the rear-right tire. |
| Mz | Self-aligning torque (vector that includes the four wheels). |
| Dy_fl | Lateral friction coefficient for the front-left tire. |
| Dy_fr | Lateral friction coefficient for the front-right tire. |
| Dy_rl | Lateral friction coefficient for the rear-left tire. |
| Dy_rr | Lateral friction coefficient for the rear-right tire. |
| dynamic_pressure | Current dynamic pressure on the tires. |
| tyre_loaded_radius | Loaded radius of each tire. |
| tyres_load | Load on each tire. |
| NdSlip | Dimensionless lateral slip friction for each tire. |
| SuspensionTravel | Suspension travel distance for each wheel. |
| CamberRad | Camber angle in radians for each wheel. |
| wheel_speed_fl | Angular speed of the front-left wheel. |
| wheel_speed_fr | Angular speed of the front-right wheel. |
| wheel_speed_rl | Angular speed of the rear-left wheel. |
| wheel_speed_rr | Angular speed of the rear-right wheel. |
| fl_tire_temperature_core | Core temperature of the front-left tire. |
| fr_tire_temperature_core | Core temperature of the front-right tire. |
| rl_tire_temperature_core | Core temperature of the rear-left tire. |
| rr_tire_temperature_core | Core temperature of the rear-right tire. |
| fl_damper_linear_potentiometer | Linear potentiometer reading for the front-left damper. |
| fr_damper_linear_potentiometer | Linear potentiometer reading for the front-right damper. |
| rl_damper_linear_potentiometer | Linear potentiometer reading for the rear-left damper. |
| rr_damper_linear_potentiometer | Linear potentiometer reading for the rear-right damper. |
| fl_wheel_load | Load on the front-left wheel. |
| fr_wheel_load | Load on the front-right wheel. |
| rl_wheel_load | Load on the rear-left wheel. |
| rr_wheel_load | Load on the rear-right wheel. |
| fl_tire_pressure | Pressure of the front-left tire. |
| fr_tire_pressure | Pressure of the front-right tire. |
| rl_tire_pressure | Pressure of the rear-left tire. |
| rr_tire_pressure | Pressure of the rear-right tire. |

# G   Additional results for Q1

We evaluate lap times of human drivers and models on three tracks (Barcelona, Red Bull Ring, Monza) using F317 and GT3 cars.

For the F317 car, Figure 10 shows the lap times on the Barcelona track, Figure 11 illustrates the lap times on the Red Bull Ring track, and Figure 12 presents the lap times on the Monza track. SAC fD (our fastest model) is generally on par or better than human drivers in the F317 car. However, only the pro eSports driver matches SAC's performance, while the second human driver in our dataset is substantially slower.

For the BMW Z4 GT3 car, Figure 13 shows the lap times on the Barcelona track, Figure 14 illustrates the lap times on the Red Bull Ring track, and Figure 15 the lap times on the Monza track. In the GT3, SAC fD is consistently outperformed by expert drivers, though it remains competitive within that group.

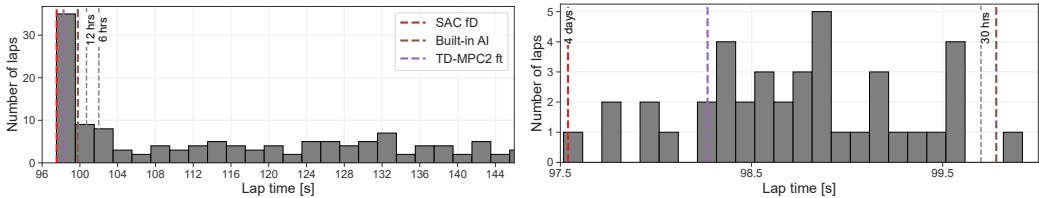

*Figure 10:* **Lap times of human drivers and algorithms on BRN with a F317 car**. ↓ Lower is better. *(left)* Overall distribution; *(right)* zoomed-in view of the best laps. Vertical lines indicate best laps by each method at end of training. SAC and TD-MPC2 outperform most human drivers with just 6 hours of training.

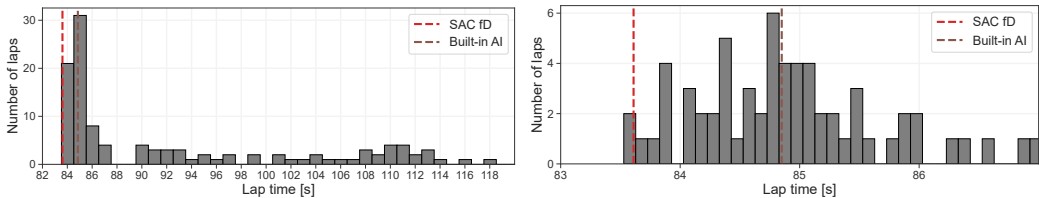

*Figure 11:* **Lap times of human drivers and algorithms on RBR with F317**. ↓ Lower is better. *(left)* Overall distribution; *(right)* zoomed-in view of the best laps.

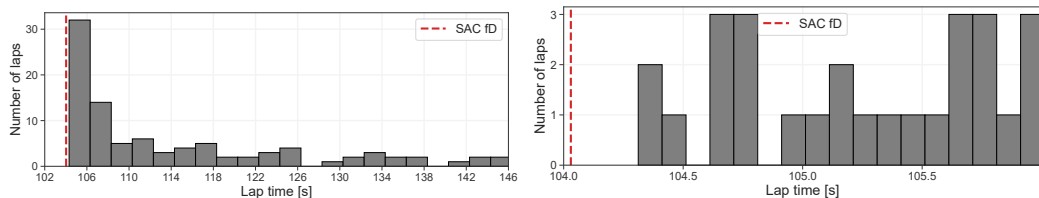

*Figure 12:* **Lap times of human drivers and algorithms on Monza with F317**. ↓ Lower is better. *(left)* Overall distribution; *(right)* zoomed-in view of the best laps.

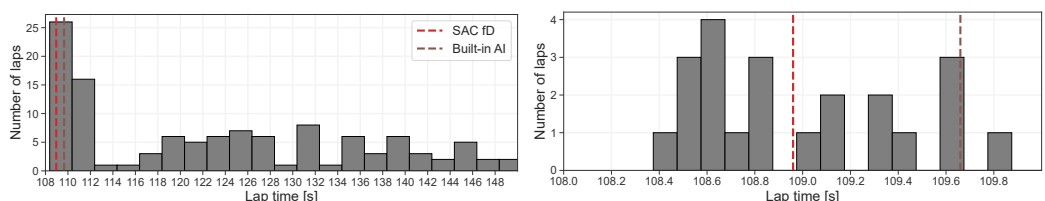

*Figure 13:* **Lap times of human drivers and algorithms on BRN with GT3**. ↓ Lower is better. *(left)* Overall distribution; *(right)* zoomed-in view of the best laps.

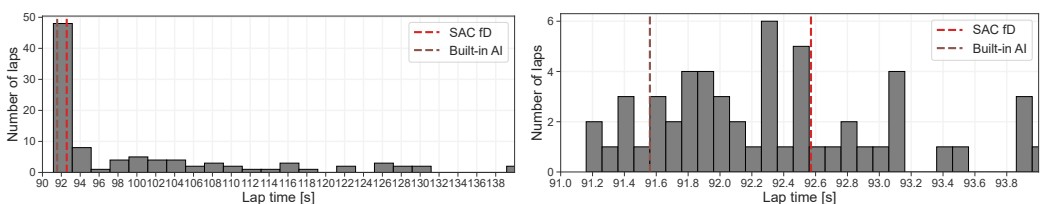

*Figure 14:* **Lap times of human drivers and algorithms on RBR with GT3**. ↓ Lower is better. *(left)* Overall distribution; *(right)* zoomed-in view of the best laps.

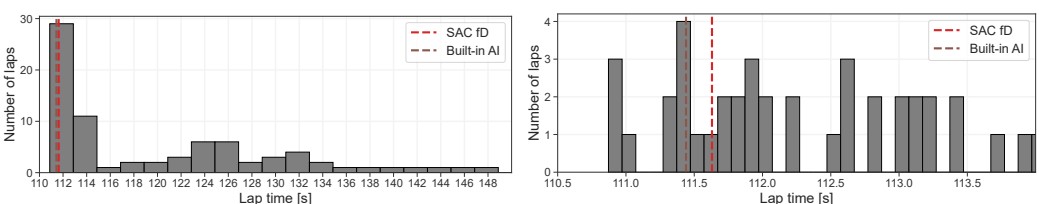

*Figure 15:* **Lap times of human drivers and algorithms on Monza with GT3**. ↓ Lower is better. *(left)* Overall distribution; *(right)* zoomed-in view of the best laps.

# H  Simulators

Figure 10 shows a comparison between different simulators, highlighting their respective strengths and weaknesses. **TORCS** (https://sourceforge.net/projects/torcs/) runs fast and is good for development, but it has poor physics and render quality. Trackmania (https://github.com/trackmania-rl/tmrl) also runs fast but has unrealistic physics. **CARLA** (https://carla.org/) runs fast but is primarily an urban cars simulator. **Learn to Race**(https://github.com/learn-to-race/l2r) offers good sensors support, but it suffers from poor physics and render quality, as well as limited tracks and cars. **Gran Turismo** (https://www.gtplanet.net/gran-turismo-7/) provides good render quality and acceptable physics, but it is not openly available and does not support custom tracks or cars and has restricted sensor support. **rFPro AVL VSM** (https://rfpro.com/) (https://www.avlracetech.com/software/) is considered one of the most realistic simulator on the market but comes with a very expensive license. **Assetto Corsa Gym** has good physics and render quality, but it has restricted sensor support which can be mitigated by creating ad hoc scripts that use information from the simulator (distance from the borders and opponents, or the 3d model of the track) with or without ROS2 interfaces.

*Table 10:* **Comparison of Simulators.** Key attributes of various driving simulators including TORCS, Trackmania, CARLA, Learn_to_Race, Gran Turismo, rFPro AVL VSM, and Assetto Corsa Gym (ours). Driver Experience Fidelity refers to the accuracy and immersion that replicate real-world driving experiences, such as the field of view, motion platform, and the realism of the pedals and steering wheel.

| Attributes | TORCS | Trackmania | CARLA | Learn_to_Race | Gran Turismo | rFPro AVL VSM | Assetto Corsa Gym |
|---|---|---|---|---|---|---|---|
| Access | Open Source | Free | Open Source | Academic license | Not Available | Expensive License | One time purchase |
| Mantained | ✗ | ✓ | ✓ | ✗ | ✓ | ✓ | ✓ |
| Simulation speed | Fast | Fast | Fast | Real time | Real time | Real time | Real time |
| Physics | Poor | Very poor | Poor | Poor | Good | Professional grade | Excellent |
| Realistic tracks | Acceptable | Bad | Bad | Acceptable | Laser scanned | Laser scanned | Laser scanned |
| Cars | >20 | 1 | 1 | 1 | >500 | 1 | 178 |
| Tracks | >20 | >25 | 1 | 3 | 37 | >20 | 19 |
| Custom tracks/cars | ✓ | ✓ | ✓ | ✗ | ✗ | ✓ | ✓ |
| Driver Experience Fidelity | Poor | Poor | Poor | Poor | Good | Professional grade | Excellent |
| ROS | ✓ | ✗ | ✓ | ✗ | ✗ | ✗ | ✓ |
| OpenAI Gym | ✓ | ✓ | ✓ | ✓ | ✓ | ✗ | ✓ |

