# OpenReview forum: "A Simulation Benchmark for Autonomous Racing with Large-Scale Human Data"
_NeurIPS.cc/2024/Datasets_and_Benchmarks_Track — NeurIPS 2024 Track Datasets and Benchmarks Poster_

### Official Review · Reviewer_Yg25 · 2024-07-20
**Review of Submission251**

**Rating:** 6
**Confidence:** 4
**Correctness:** the dataset is constructed in a sound…
**Clarity:** The paper is well-written and clear.

**Review:**

**Quality**

The quality of this paper is good. I can see a lot of effort in building the platform, collecting the data, and training the agents. The results analysis also provides some interesting comparisons and conclusions.

**Clarity**

The paper is well-written and clear.

**Originality**

The simulation interfaces and human demonstration datasets are originally proposed in this paper. In general, I think these contents are original and new.

**Significance**

The racing simulation seems an interesting topic. However, the authors do not emphasize the impact of this work on the broad community. In addition, it is not clear what the differences are between this one and existing ones.

**Strengths:**

In general, the paper writing is clear. The contribution of the platform and datasets has high quality.

**Additional Feedback:**

N/A

**Documentation:**

The details on data collection and organization are available. The dataset URL is also avaliable

**Limitations:**

The authors honestly discuss a lot of limitations of this work, which I appreciate. Some of them can be improved but others cannot. For example, the real-time simulation seems a limitation of the AC engine, which may pose the motivation for using this platform. The experiment with image observations and the control of shifts seems achievable things, which I hope the authors can address in the future.

Additional limitations from my side are that this paper does not argue the positive impact of this platform and research on the autonomous driving of civil vehicles. I understand that the racing car itself is an interesting domain, but it doesn’t show the contribution of this platform to the racing vehicle community. The second thing is that the scenario considered in this paper only involves one car if I understand correctly. I guess the interaction between cars at high speed could be interesting, like in F1 and Touring Car Racing, which makes the RL agents able to compete with human drivers.

**Opportunities For Improvement:**

1.	A table for comparing racing simulators from different perspectives would be helpful. For example, Learn-to-Race has 3 tracks and this paper has 4 tracks. It seems not a big improvement. In which aspect does the provided simulator outperform Learn-to-Race?
2.	The results in Figure 6 seem interesting. The type of car influences the results a lot. Could the authors provide potential explanations?
3.	I don’t get the point of this sentence: “However, this real-time constraint is beneficial for testing algorithms in real-world conditions.” Is it always better to have a faster simulation? Why is it beneficial for testing? The only thing I can think about is that human reaction requires a real-time simulation. However, it is only required during the data collection.
4.	Do the trained agents consider the interaction with other cars?
5.	In Figure 4, there are overlaps between text.
6.	In Line 149, all ows -> allows

**Relation To Prior Work:**

The related work section seems thorough but not very intuitive. A table for comparing racing simulators from different perspectives would be helpful.

**Summary And Contributions:**

This paper is motivated by the high costs of racing vehicle acquisition and management as well as the limited physics accuracy of open-source simulators. The authors propose a racing simulation platform based on the game Assetto Corsa to test, validate, and benchmark autonomous driving algorithms in realistic and challenging scenarios. In summary, the contributions are (1) the development of this simulation platform; (2) several state-of-the-art algorithms tailored to the racing environment; (3) a comprehensive dataset collected from human drivers.

---

> ### Author Rebuttal · Authors · 2024-08-16
>
> We thank the reviewer for their valuable feedback. We address your comments in the following.
>
> ***Q:*** The racing simulation seems an interesting topic. However, the authors do not emphasize the impact of this work on the broad community.
>
> ***A:*** Our research offers significant contributions to the NeurIPS community and the broader field of machine learning. By introducing a new benchmark and an inexpensive application, our work provides valuable resources for researchers. The algorithms developed and refined in high-speed, high-stress racing environments can enhance various aspects of machine learning, such as improving planning and safety mechanisms. These can be applied beyond autonomous driving, offering potential improvements in diverse areas such as robotics and real-time decision making.
>
> ---
>
> ***Q:*** this paper does not argue the positive impact of this platform and research on the autonomous driving of civil vehicles. I understand that the racing car itself is an interesting domain, but it doesn’t show the contribution of this platform to the racing vehicle community.
>
> ***A:*** As already mentioned in the reply to Q2 of reviewer 7WzT, there are several mods to add urban roads/cities and urban cars. This ensures our research can be effectively translated to improve autonomous driving systems in complex urban settings.
>
> Assetto Corsa's advanced physics engine provides a realistic testing ground, ensuring that performance gains in simulation translate effectively to real-world conditions
>
> ---
>
> ***Q:*** It is not clear what the differences are between this one and existing ones. A table for comparing racing simulators from different perspectives would be helpful. For example, Learn-to-Race has 3 tracks and this paper has 4 tracks. It seems not a big improvement. In which aspect does the provided simulator outperform Learn-to-Race?
>
> ***A:*** We appreciate the suggestion. To address this, we have included a table (see PDF in the author's rebuttal) that compares our simulator with state-of-the-art racing simulators, including Learn-to-Race.
>
> ---
>
> ***Q:*** The results in Figure 6 seem interesting. The type of car influences the results a lot. Could the authors provide potential explanations?
>
> ***A:*** This surprised us as well. The reason could lie in the automatic gearbox used by our model, which is provided by AC. The professional driver noted that while the automatic gearbox in the F317 is good enough, it switches gears at very low RPMs when driving the GT3 car, which is suboptimal. To address this, we plan to investigate models that can support a mix of discrete and continuous action spaces, allowing the model to make better decisions on when to shift gears. This approach should improve performance across different car types.
>
> ---
>
> ***Q:*** I don’t get the point of this sentence: “However, this real-time constraint is beneficial for testing algorithms in real-world conditions.” Is it always better to have a faster simulation? Why is it beneficial for testing? The only thing I can think about is that human reaction requires a real-time simulation. However, it is only required during the data collection.
>
> ***A:*** We intended to convey that while the real-time constraint might be seen as a limitation, it can also be beneficial as it is closer to real-life car behavior, providing a more realistic and rigorous testing environment. There is a trade-off between simulation fidelity and run time; higher complexity increases simulation time. To address this limitation, our platform enables running multiple instances simultaneously. This approach ensures that we can maintain high simulation fidelity while efficiently conducting large-scale experiments.
>
> ---
>
> ***Q:*** Do the trained agents consider the interaction with other cars?
>
> ***Q:*** The scenario considered in this paper only involves one car if I understand correctly. I guess the interaction between cars at high speed could be interesting, like in F1 and Touring Car Racing, which makes the RL agents able to compete with human drivers.
>
> ***A:*** Indeed, this is a promising direction for research. However, we do not address this in the current work, which focuses on the single car scenario (time attack). Our platform can handle interactions with other cars, and we plan to explore this aspect in future research.
>
> ---
>
> ***Q:*** In Figure 4, there are overlaps between text.
>
> ***Q:*** In Line 149, all ows -> allows
>
> ***A:*** Thanks. We have fixed these issues.
>
> ---
>
> Please do not hesitate to let us know if you have any additional comments.

---

### Official Review · Reviewer_rG1V · 2024-07-25
**Simulator for high-fidelity racing**

**Rating:** 6
**Confidence:** 4
**Correctness:** The dataset and benchmark are sound.
**Clarity:** Yes, the paper is easy to read

**Review:**

**Strength:**

1. This is the first open-sourced platform for high-speed racing. Given the realistic physics, research conducted on this platform could benefit real-world deployment. The high-fidelity vehicle dynamics allow researchers to explore how to push the vehicle performance to the limit, i.e. using manual shifting instead of the autoshifing.
2. The paper is well-written. The code, data, and website are well-prepared.
3. The experiments prove the completeness whole system and show interesting results

**Weakness:**

1. As there are human participants, I believe IRB approval would be required no matter whether they are volunteering or get-paid. But I can find it with the submission.
2. Though it is said in the abstract that *"algorithms are evaluated in offline RL settings"*, I can not find any offline RL methods in the experiments. There are only naive SAC and TD-MPC2 whose buffer is filled with offline data initially. I think some SOTA offline-RL methods can be benchmarked as well like TD3+BD, IQL, and so on. This is important as the simulator runs in real-time and data-collection is inefficient and thus could be a good plyground for offline RL.
3. As acknowledged in the limitation, the lack of multi-modal observation (semantic/RGB/depth camera) will limit the possibility of this simulator. For example, the experiments show that without the pre-calculated reference path, the agent finds it hard to learn a good policy and even fails. However, such a path is hard to calculate in real-world deployment according to the paper, so it is important to study how to drive without reference. In this case, image input can be useful as it provides comprehensive information about the weather/road surface/road structure/elevation changes, which allows the agent to infer such a reference path implicitly. Thus, I suggest to have the cameras provided with this platform.

**Strengths:**

See above

**Additional Feedback:**

N/A

**Documentation:**

Yes, the dataset and simulator have sufficient explanation and guidance.

**Ethics:**

The research involved human subjects but without IRB approval provided.

**Opportunities For Improvement:**

New experiment baselines are needed.

**Relation To Prior Work:**

The related work section is complete

**Summary And Contributions:**

This paper proposes an open-sourced platform that allows researchers to study racing strategies. The simulator is based on, as far as I know, the best driving game in terms of the fidelity of vehicle dynamics, which makes the research output with this simulator feasible to be transferred to the real world. As the simulator can only run in real-time, making the data collection hard, a dataset consisting of human demonstration and RL agent experience is shipped with this work. As a result, both online and offline learning methods can be studied with this platform.

---

> ### Author Rebuttal · Authors · 2024-08-16
>
> We thank the reviewer for their valuable feedback. We address your comments in the following.
>
>
>
>
> ***Q:*** As there are human participants, I believe IRB approval would be required no matter whether they are volunteering or get-paid. But I can find it with the submission.
>
> ***A:*** After reviewing the relevant guidelines, we concluded that IRB approval was not needed for this study. This conclusion was confirmed by Ethics Reviewer 3Rtg. However, we acknowledge the importance of addressing other ethical considerations. Specifically, we have obtained explicit consent from all participants involved in the study. Detailed information on these measures has been added to the manuscript to provide clarity and transparency.
>
> ---
>
> ***Q:*** Though it is said in the abstract that "algorithms are evaluated in offline RL settings", I can not find any offline RL methods in the experiments. There are only naive SAC and TD-MPC2 whose buffer is filled with offline data initially. I think some SOTA offline-RL methods can be benchmarked as well like TD3+BD, IQL, and so on. This is important as the simulator runs in real-time and data-collection is inefficient and thus could be a good plyground for offline RL
>
> ***Q:*** Opportunities For Improvement: New experiment baselines are needed.
>
> ***A:*** We agree that this could be worded better. In the current work, we are conducting offline-to-online RL in the track transfer experiments. We acknowledge the importance of including pure offline RL baselines and are committed to benchmarking state-of-the-art offline RL methods such as TD3+BC and IQL for the camera-ready version.
>
> ---
>
> ***Q:*** … image input can be useful as it provides comprehensive information about the weather/road surface/road structure/elevation changes, which allows the agent to infer such a reference path implicitly. Thus, I suggest to have the cameras provided with this platform.
>
> ***A:*** We agree that vision-based input is crucial as it contains rich information. We are currently planning to integrate cameras into the platform and have already developed a running prototype. Additionally, by making it open source, we invite everyone to contribute and add further features on perception and other capabilities.
>
> ---
>
> Please do not hesitate to let us know if you have any additional comments.

---

### Official Review · Reviewer_7WzT · 2024-07-28
**Comprehensive work with a few minor limitations.**

**Rating:** 8
**Confidence:** 4

**Review:**

This paper makes a significant contribution to autonomous racing research, providing a comprehensive approach that combines rigorous experimentation and comprehensive data. The inclusion of open-source resources enhances the paper's impact and facilitates further research. Overall, the paper is a valuable addition to the field, offering insights and resources that will likely inform future work in autonomous racing and related areas.

**Strengths:**

The paper's strengths include its comprehensive simulation benchmark for autonomous racing, large-scale dataset, and rigorous experimental evaluation. The authors provide open-source code and data, ensuring transparency and reproducibility. Clear writing effectively communicates their ideas, and strong motivation highlights the significance of autonomous racing research. Interdisciplinary relevance and consideration of ethical implications further enhance the paper's quality, making it a valuable contribution to the field of autonomous racing and reinforcement learning.

**Additional Feedback:**

It will be good to extend the scope  to include sensor level simulation for various sensing modality.

**Clarity:**

The paper is well-written, with clear and concise language, proper structure, and good organization, effectively communicating the authors' ideas, methods, and results. The writing is strong, with a clear introduction, concise background information, transparent methodology, and effective presentation of results, making the paper easy to follow and understand. The paper is overall well-crafted and effectively conveys the authors' research.

**Correctness:**

The claims made in the submission appear to be correct, and the dataset is constructed in a sound way, with the authors providing a comprehensive simulation benchmark for autonomous racing. While the submission itself may not contain exhaustive implementation details, the supplementary material and open-sourced code serve as a reference, offering transparency and facilitating reproducibility. The authors' evaluation methods and experimental design seem appropriate, covering various aspects such as track layouts, vehicle dynamics, and control algorithms, and utilizing suitable metrics. Although a deeper examination of the source code and supplementary material would provide further insight, the submission appears rigorous and well-constructed, with the authors demonstrating a commitment to openness and reproducibility by making their code and dataset publicly available.

**Documentation:**

This is one of the strengths of the paper. There is great level of details available both in the paper and supplementary material.

**Ethics:**

No major concerns other than the lack of visibility on the skill level and driving behaviors of the human drivers in the study.

**Limitations:**

Already mentioned in the opportunities for improvement/

**Opportunities For Improvement:**

The work has limitations in its significance, relevance, and quality. It focuses on a specific simulator (Assetto Corsa) and racing context, limiting generalizability, and builds upon existing research without introducing entirely new concepts. The experimental evaluation, although rigorous, is limited to specific algorithms and scenarios, and the choice of metrics and evaluation protocols may not fully capture the complexity of autonomous racing. Additionally, the work primarily focuses on reinforcement learning and model predictive control, leaving other approaches unexplored, and its interdisciplinary relevance is mainly limited to machine learning, control theory, and computer vision. Furthermore, the consideration of ethical and social implications is brief, and the potential negative societal impacts, such as job displacement or increased risk of accidents, are not fully explored.

**Relation To Prior Work:**

Section 2 cites all the relevant work and seamlessly flows into the problem definition section.

**Summary And Contributions:**

This study presents a cutting-edge testing framework for self-driving cars using the Assetto Corsa simulator, facilitating assessment of innovative control methods, including reinforcement learning (RL) and model predictive control (MPC), in complex and realistic driving scenarios. The experiments reveal that artificial intelligence (AI) techniques can match, and even surpass, human driving performance, leveraging expert demonstrations, rapid adaptation, and strategic exploration. Notably, the results show that RL can achieve lap times comparable to or better than human drivers, and benefits from human demonstrations, few-shot transfer, and learning to race without a predefined reference path. Overall, I'm impressed by the paper's comprehensive approach, thorough evaluation, and meaningful contributions, which include a valuable dataset of human driving data and RL policy data, as well as open-source code, significantly advancing the field of autonomous racing research.

---

> ### Author Rebuttal · Authors · 2024-08-16
>
> We thank the reviewer for their valuable feedback. We address your comments in the following.
>
> ***Q:*** No major concerns other than the lack of visibility on the skill level and driving behaviors of the human drivers in the study.
>
> ***A:*** We acknowledge the importance of understanding the skill level and driving behaviors of the human drivers in our study. While Ethics Reviewer 3Rtg noted that this is not an ethical concern, we recognize that additional context can enhance the robustness of our findings. Due to the constraints of our study design and the scope of data collection, we were unable to provide detailed individual assessments of each driver's skill level and behavior. However, we categorized participants into four distinct groups based on their self-reported driving experience: One professional sim driver, four experts who regularly train and compete online, five casual drivers with some experience, and five beginners driving on a racing simulator for the first time. We will make the categorization more explicit in the camera-ready version
>
> ---
>
> ***Q:*** Opportunities For Improvement: The work has limitations in its significance, relevance, and quality. It focuses on a specific simulator (Assetto Corsa) and racing context, limiting generalizability, and builds upon existing research without introducing entirely new concepts
>
> ***Q:*** the work primarily focuses on reinforcement learning and model predictive control, leaving other approaches unexplored, and its interdisciplinary relevance is mainly limited to machine learning, control theory, and computer vision.
>
> ***A:*** We acknowledge that our study focuses on the Assetto Corsa simulator and a racing context, which may limit generalizability. However, Assetto Corsa was chosen due to its realistic physics engine and its relevance to racing research. Future work could indeed explore more diverse driving environments to enhance generalizability. Additionally, Assetto Corsa can be used with urban road maps and urban cars through mods, including AIs driving urban cars in different weather conditions such as night, fog, and wet environments. This highlights that our current study serves as a baseline for a more extended and complete study of AI and classical approaches for autonomous racing. Moreover, it sets the foundation for a platform that will be extended with sensor simulation, multi-vehicle online scenarios, and urban environments.
>
> While our work builds on existing research, it introduces novel applications of RL in the context of autonomous racing, particularly in how these methods can benefit from human demonstrations and adapt to complex scenarios. We aim to expand on these innovations in future research. Our primary focus was on RL and MPC due to their prominence and potential in autonomous racing. We recognize the value of exploring additional approaches and will consider integrating other methodologies in future work to provide a more holistic view, especially driving from vision.
>
> ---
>
> Please do not hesitate to let us know if you have any additional comments.

---

### Author Rebuttal · Authors · 2024-08-16

We thank all reviewers for their thoughtful comments and for their constructive feedback. We have responded to your individual comments. We believe that all comments have been addressed, but are happy to address any further comments from reviewers.

As suggested by Reviewer Yg25, we have included a PDF in the author rebuttal that contains a table comparing our simulator with state-of-the-art racing simulators

Best,

Authors of “A Simulation Benchmark for Autonomous Racing with Large-Scale Human Data” (submission 251)

---

### Decision · Program_Chairs · 2024-09-26

**Decision:**

Accept (Poster)

**Comment:**

This work built a racing simulation benchmark on top of the Assetto Corsa video games. Several RL and MPC algorithms are evaluated. Human driving data is also collected

After rebuttal, this submission received 8, 6, 6, so there was a consensus of acceptance. AC has checked the submission, the reviews, and the rebuttal. Many prior works have explored the RL for car racing in video games, such as Sony Gran Turismo Sophy. In that sense, the novelty of this work is limited. However, AC believes the open-source data and source code will be useful for the community and the authors executed the work very well. Thus, acceptance is recommended.